# Identification of an Immune-Related Prognostic Signature for Glioblastoma by Comprehensive Bioinformatics and Experimental Analyses

**DOI:** 10.3390/cells11193000

**Published:** 2022-09-26

**Authors:** Shengda Ye, Bin Yang, Tingbao Zhang, Wei Wei, Zhiqiang Li, Jincao Chen, Xiang Li

**Affiliations:** Department of Neurosurgery, Zhongnan Hospital of Wuhan University, Wuhan 430071, China

**Keywords:** immune-related prognostic signature, glioblastoma, bioinformatics, experiment, nomogram, prognosis, WGCNA

## Abstract

Background: Glioblastoma (GBM), which has a poor prognosis, accounts for 31% of all cancers in the brain and central nervous system. There is a paucity of research on prognostic indicators associated with the tumor immune microenvironment in GBM patients. Accurate tools for risk assessment of GBM patients are urgently needed. Methods: In this study, we used weighted gene co-expression network analysis (WGCNA) and differentially expressed gene (DEG) methods to screen out GBM-related genes among immune-related genes (IRGs). Then, we used survival analysis and Cox regression analysis to identify prognostic genes among the GBM-related genes to further establish a risk signature, which was validated using methods including ROC analysis, stratification analysis, protein expression level validation (HPA), gene expression level validation based on public cohorts, and RT-qPCR. In order to provide clinicians with a useful tool to predict survival, a nomogram based on an assessment of IRGs and clinicopathological features was constructed and further validated using DCA, time-dependent ROC curve, etc. Results: Three immune-related genes were found: PPP4C (*p* < 0.001, HR = 0.514), C5AR1 (*p* < 0.001, HR = 1.215), and IL-10 (*p* < 0.001, HR = 1.047). An immune-related prognostic signature (IPS) was built to calculate risk scores for GBM patients; patients classified into different risk groups had significant differences in survival (*p* = 0.006). Then, we constructed a nomogram based on an assessment of the IRG-based signature, which was validated as a potential prediction tool for GBM survival rates, showing greater accuracy than the nomogram without the IPS when predicting 1-year (0.35 < *Pt* < 0.50), 3-year (0.65 < *Pt* < 0.80), and 5-year (0.65 < *Pt* < 0.80) survival. Conclusions: In conclusion, we integrated bioinformatics and experimental approaches to construct an IPS and a nomogram based on IPS for predicting GBM prognosis. The signature showed strong potential for prognostic prediction and could help in developing more precise diagnostic approaches and treatments for GBM.

## 1. Introduction

Gliomas account for 31% of all cancers in the brain and central nervous system (CNS), and 81% of malignant brain and CNS tumors diagnosed in the United States [1]. The World Health Organization’s (WHO) grading criteria are used to classify these malignancies. Among the gliomas, GBM is the most common type of primary malignant brain tumor, accounting for 16% of all primary brain and CNS malignancies [2]. The 5-year survival rate is only 3.3%. The average incidence of the disease is 3.19/100,000, and the median age at the time of diagnosis is 64 years old [2]. Today, the basic treatment approach is still surgery, combined with radiotherapy, chemotherapy, and other comprehensive treatment methods [3]. Even with the progress made in surgical resection, GBM patients still have a poor prognosis, with median survival of 15 months [1,3].

As a result of the poor prognosis for glioblastoma, novel prognostic biomarkers are urgently required. Immunotherapy is a sort of cancer treatment that uses the immune system to combat cancer [4]. However, immunotherapy has made little progress in glioblastoma over the past 10 years because of the intracranial location and heterogeneity of tumors, and the distinct immunosuppressive tumor microenvironment. New prognostic biomarkers should be developed [5]. In recent years, continuous progress in bioinformatics has promoted the use of public database mining for identifying cancer biomarkers. With the deepening of theoretical research, IRGs (immune-related genes) have shown promise in immunotherapy for a variety of cancers [6,7]. Fu et al. explored immune-related prognostic markers of prostate cancer and their possible mechanisms, providing a basis for individualized diagnosis and treatment [8]. Tao et al. found immune-related prognostic biomarkers that could strengthen immunotherapy efficacy. Using IRGs to screen for prognostic biomarkers has undoubtedly become a hub of tumor immunotherapy research [6,9]. There is a paucity of research on prognostic indicators associated with prognosis in GBM patients; the research is still limited to a few special genes, and research on its mechanism is lacking. Our research is intended to re-establish a novel IRG-based prognostic model. Comprehensively including more gene targets and combining molecular drugs may be important in the treatment of GBM. Thus, we attempted to construct an immune-related prognostic biomarker that could be verified as an independent biomarker that could guide appropriate treatment for improved outcomes in GBM.

## 2. Materials and Methods

### 2.1. GBM Tissue Specimen Collection

Five specimens of surgically removed GBM tissues were collected from the Department of Neurosurgery of Zhongnan Hospital of Wuhan University from December 2021 to June 2022, and six specimens of normal peritumoral tissues were collected. All patients had a pathologically confirmed GBM diagnosis and signed an informed consent form, and the study was approved by the ethics committee.

### 2.2. Collection of Datasets and Immune-Associated Genes

Figure 1 depicts the research steps of this study, indicating the recognition and authentication of immune-associated biomarkers for GBM prognosis. The Cancer Genome Atlas (TCGA) database (https://genomecancer.ucsc.edu/ accessed on 10 May 2022) was used to obtain the fragments per kilobase per million mapped reads (FPKM) standardized data of GBM. We excluded tumor samples that did not have enough clinical information for further analysis. In total, 167 GBMs with complete survival data and underlying clinical variables (age, tumor, gender, grade, and stage) and 5 normal samples were used in the present study. The R package DEseq.2 [10] was used to perform TCGA-GBM data normalization (including library-size normalization and log transformation).

In addition, 5 independent data cohorts (GSE15824 [11], GSE4290 [12], GSE51062 [13], GSE2817 [14], and GSE4412 [15]) were collected through the Gene Expression Omnibus (GEO) database (http://www.ncbi.nlm.nih.gov/geo/ accessed on 10 May 2022). We merged GSE15824, GSE4290, and GSE51062 because they all used the same application platform (GPL570). First, we downloaded the original data for the 3 datasets and used the R package affy [16] to perform RMA normalization on the original data. In addition, we used the R package insilico-Merging to preprocess, merge, and adjust the 3 datasets. We then used the GPL570 annotation file probes to match probe IDs to gene symbols. MMD1 was created by combining 28 normal tissues with 151 GBM tissues, which was used for differentially expressed gene (DEG) identification. Furthermore, GSE2817, including 25 GBMs with complete survival information, and GSE4412, including 85 GBMs with complete survival information, were merged as MMD2 (n = 110), and was used for prognostic value identification.

The IRGs came from the ImmPort database (https://www.import.org/ accessed on 10 May 2022). We collected 2498 IRGs from the database and selected 1617 genes that coincided with the TCGA-GBM gene list for further investigation.

### 2.3. WGCNA to Filter Key Module

An expression matrix of the 1617 IRGs obtained from TCGA-GBM data was constructed using 2 methods, the sample network method and the goodSamplesGenes function. Exceptional values were also discovered using the cut-off Z.Ku < −2.5 (Z.ku = (ku-mean(k))/(sqrt(var(k)))). Unqualified samples were not included in the analysis, and we obtained 170 samples. Next, we established co-expression networks through the R package WGCNA [17], and the IRGs were divided into gene modules by the branch cutting method [18]. Some important parameters were set in the branch cutting method, including minClusterSize = 30 and deepSplit = 2. After segmenting the IRGs, a line segment with a correlation of 0.76 was chosen to join the highly correlated modules, which was carried out by examining the difference in module eigengenes. We initially assessed gene significance (GS) to look for hub modules linked to disease status (the chosen illness characteristic is GBM or normal). In addition, GS was used to compute module significance (MS). MS was calculated as the average GS of all genes in this module. Following the steps outlined above, we selected the most relevant module, which was estimated to be the most important module. In addition, genes with |cor.geneModuleMembership| > 0.8 and |cor.geneTraitSignificance| > 0.2 were also considered as key genes in WGCNA and were subsequently collected.

### 2.4. Identification of Differentially Expressed Immune-Related Genes

Based on MMD1, with 151 GBMs and 28 normal samples, we identified DEGs from GBMs and normal tissues using the R package limma [19]. DEGs were discovered using the expression matrix of 1242 IRGs in MMD1 based on limma, and those with adjusted *p*-value < 0.05 and |log2FC| ≥ 1.0 were further analyzed.

### 2.5. Identification of the Hub Gene

Genes that were discovered in both WGCNA and DEG analysis were designated as hub genes for future study.

### 2.6. Potential Prognostic Gene Identification

After identifying overlapping hub genes in WGCNA and DEG analysis, we tried to filter out potential prognostic biomarkers. The R package survival [20] was used to generate 2 independent survival analyses: overall survival (OS) and disease-specific survival (DSS). The cut-off criteria was *p* < 0.05. Genes that showed significant values were considered as potential prognostic genes in both survival analyses. OS was further subjected to univariate Cox analysis for these potential prognostic biomarkers. The potential prognostic genes were functionally annotated by the R package clusterProfiler [21] for KEGG pathway and GO enrichment analyses. To define significant BPs and KEGG pathway terms, we used *p* < 0.05 as the criterion.

### 2.7. Transcription-Level Expression Validation by RT-qPCR

Total RNA was extracted from GBM tumors and normal peritumoral tissues and quantified by a nanophotometer (Implen GmbH, München, Germany) using RNAiso Plus (Takara, Kusatsu, Shiga, Japan). Absorbance ratios of 260/280 and 260/230 nm were calculated to assess RNA quality. HiScript^®^ III RT SuperMix (Vazyme, Nanjing, China) and ChamQ Universal SYBR qPCR Master Mix (Vazyme, Nanjing, China) were used for RT-qPCR with 500 ng total RNA according to the kit’s instructions. In order to determine the relative mRNA expression level, we employed the 2(−Ct) technique. Appendix A lists the primers used in this investigation.

### 2.8. Validation of Internal Expression Level of Potential Prognostic Genes

After screening out potential prognostic genes, we validated the difference in the transcription expression levels of these genes between GBM and normal tissues by using the Gene Expression Profiling Interactive Analysis (GEPIA) webtool (http://gepia.cancer-pku.cn/ accessed on 14 May 2022) [22]. We also explored the associations among these genes using this webtool. In addition, translation-level expression differences between normal samples and GBMs were acquired from the Human Protein Atlas (HPA) database (https://www.proteinatlas.org/ accessed on 14 May 2022).

### 2.9. Establishing a Prognostic Risk System

We investigated the prognostic value of the potential prognostic genes by combining the regression coefficients (Coef) of prognostic biomarkers in univariate Cox analysis of OS with the prognostic biomarker expression levels and established a prognostic risk assessment system. The GBM sample risk score (RS) was defined as follows:$$\mathrm{Risk}\;\mathrm{score}=\sum_{i=1}^{n}{\mathrm{Coef}}_{i}\times {\mathrm{Exp}}_{i}$$

In the equation, Coef represents the regression coefficient, which is determined by multivariate Cox regression analysis, and Exp represents the expression level of the prognostic biomarker. The risk system’s predictive value was measured using the RSs of GBM samples derived from TCGA-GBM data and MMD2. The GBM samples in all datasets were split into 2 groups (low-risk and high-risk) based on the median RS in each dataset. We used the R package survival to perform OS analysis. Additionally, the R package survivalROC [23] was used to display time-dependent (1-, 3-, and 5-year) receiver operating characteristic (ROC) curves.

### 2.10. Analysis of Cox Proportional Hazards Regression

In order to validate the prognostic significance of the system, the risk score was evaluated by the system, and other basic clinical features (such as chemotherapy, age, and gender) in the TCGA-GBM data were selected and subjected to univariate Cox analysis of OS. Factors with *p*-value < 0.05 were screened via multivariable Cox analysis to determine whether they were independent of other clinical factors predicting OS in GBM patients. Visualization was carried out by using the R software package forestplot [24].

### 2.11. Creation and Validation of Nomogram

To better understand the clinical use of this risk system, we attempted to establish a nomogram. First, we performed cross-validation to avoid overfitting, and using the obtained immune-related prognostic risk system, the nomogram was constructed by using the R package rms. A calibration curve was then drawn to examine the nomogram. Correspondingly, the best prediction potential was the 45° line. Furthermore, we carried out a decision curve analysis using the R package rmda [25] to investigate the nomogram’s clinical utility.

### 2.12. Functional Exploration of Prognostic Risk System

Gene set enrichment analysis can be beneficial for clarifying the role of a prognostic risk system in biological behavior. From the TCGA-GBM data, we first calculated the risk score system’s median value. Then, the 167 GBMs were divided into 2 groups: low-expression and high-expression. The annotated gene set was c2.cp.kegg.v7.4.symbols.gmt. Once this work was carried out, only FDR < 25% and gene size n ≥ 20% were found to be significantly enriched by GSEA (http://software.broadinstitute.org/gsea/index.jsp/ accessed on 14 May 2022) [26] and KEGG pathways, with *p* < 0.05, |ES| > 6.

### 2.13. Association between Immune Cells and Hub Gene Expression

It is recognized that immune cells may be independent predictors of cancer survival. Therefore, the relationship between prognostic biomarkers and immunocytes was obtained through TIMER (https://cistrome.shinyapps.io/timer/ accessed on 16 May 2022) [27]. Factors with *p*-value < 0.05 and |correlation coefficient (cor)| ≥ 0.2 were deemed highly related to the degree of immune cell infiltration, as previously noted. We also used the Cell Type Identification via Estimating Relative Subsets of RNA Transcripts (CIBERSORT) (https://cibersort.stanford.edu/ accessed on 16 May 2022) [28] to assess the interaction between immunocytes and the immune-related prognostic signature.

### 2.14. Statistical Analysis

All the data were analyzed and plotted using the R software (version 4.0.2). The survival curve was plotted using the log-rank test, the Wilcoxon rank-sum test was used to assess differential expression, and Student’s *t*-test was applied to analyze continuous variables, with *p* < 0.05 (two-tailed) considered statistically significant.

## 3. Results

### 3.1. Identification of Key Module

After removing 5 outliers, 162 GBMs were included in WGCNA (Figure 2A). The soft threshold (β) = 3 (slope = −1.28) was set to evaluate adjacency (Figure 2B). IRGs were then identified and divided into gene modules. As seen in Figure 2C, 11 modules were discovered. Genes with low correlation with the interest trait were classified into gray modules, which were not included in subsequent analysis. Among the modules, four modules showed a significant association with disease status (normal or GBM): blue (*p* = 1 × 10^−32^, R^2^ = 0.76), black (*p* = 4 × 10^−^^7^, R^2^ = −0.38), turquoise (*p* = 1 × 10^−^^5^, R^2^ = −0.33), and purple (*p* = 2 × 10^−^^4^, R^2^ = −0.28) (Figure 2D). Among the 10 modules, the blue module had the strongest correlation with disease status, and the GS and MM of the blue module were significantly correlated (cor = 0.81, *p* = 1.7 × 10^−51^), as shown in Figure 2E. We also explored the relationship between MM and GS in the black, turquoise, and purple modules, and the results were meaningful: black (cor = 0.71, *p* = 2.8 × 10^−12^), purple (cor = 0.39, *p* = 0.023), and turquoise (cor = 0.45, *p* = 1.8 × 10^−14^). As shown in Figure 2I, the MS of the four modules was the highest among the 10 modules. Therefore, we had reason to think that those four were key modules. The network heatmap of these IRGs is shown in Figure 2J. The classic MDS diagram (Figure 2K) shows that the 10 modules are independent of each other.

### 3.2. Hub Gene Screening

According to the set cut-off criteria, 209 DEGs (121 overexpressed and 87 underexpressed) were screened using MMD1 (Figure 3A). We also drew a heatmap to show the expression differences of DEGs between normal and tumor tissues (Figure 2B). In addition, 86 genes were identified in WGCNA. Finally, 29 overlapping hub genes between TCGA-GBM-based DEGs and WGCNA-based hub genes were selected for further analysis (Figure 3C).

### 3.3. Potential Prognostic Gene Screening

We subsequently obtained 29 hub genes in the total survival and disease-specific survival analyses. Only three hub genes, PPP4C, C5AR1, and IL10, showed survival difference in OS (Figure 3D). In addition, six genes, IL10, PPP4C, C5AR1, CD74, PIK3R5, and RNASE2, had *p*-values < 0.05 in DSS analysis (Figure 3E). Furthermore, three genes, C5AR1, IL10, and PPP4C, overlapping in OS and DSS analyses were identified as predictive genes. Figure 4 shows the survival curves for the three prognostic genes. GBM patients with higher C5AR1 expression showed worse OS (*p* = 0.022; Figure 4A) and DSS (*p* = 0.013; Figure 4D). Similarly, patients with higher IL10 expression had worse OS (*p* = 0.025; Figure 4B) and DSS (*p* = 0.0038; Figure 4E) than those with lower IL10 expression. Furthermore, as shown in Figure 4C, patients with high PPP4C expression had better OS than those with low expression (*p* = 0.0048). In the DSS analysis of PPP4C, the outcome matched what we found in OS analysis (*p* = 0.0068; Figure 4F). We then obtained the three prognostic genes for Cox regression analysis. The findings also revealed that C5AR1 (*p* < 0.001, HR = 1.215), IL10 (*p* < 0.001, HR = 1.047), and PPP4C (*p* < 0.001, HR = 0.514) were prognostic biomarkers for GBM (Figure 5A).

### 3.4. Potential Function of the Prognostic Biomarkers

For the purpose of investigating the functions of these prognostic biomarkers, we included the three genes in GO and KEGG enrichment analysis. As shown in Figure 5B, GO BP analysis indicated that the three prognostic biomarkers were highly enriched in adaptive immunological response, based on immunoglobulin superfamily domain-dependent somatic recombination of immune receptors, positive regulation of cytokine production, adaptive immune response, macrophage activation, lymphocyte-mediated immunity, adaptive immune response based on somatic reorganization of immune receptors formed by immunoglobulin superfamily domains, lymphocyte activation involved in immune response, regulation of immune effector process, presentation of peptide antigen, antigen processing, and T cell activation. KEGG analysis indicated that the prognostic biomarkers mainly acted in cytotoxicity mediated by natural killer cells, tuberculosis, inflammatory bowel disease, leishmaniasis, asthma, Th1 and Th2 cell differentiation, *Staphylococcus aureus* infection, African trypanosomiasis, allograft rejection, and amoebiasis (Figure 5C).

### 3.5. Multilayered Validation of Prognostic Biomarkers

For the purpose of obtaining accurate conclusions, we performed a comprehensive validation of the three prognostic biomarkers. Compared with normal tissues, C5AR1 (*p* < 0.05; Figure 6A), IL10 (*p* < 0.05; Figure 6B), and PPP4C (*p* < 0.05; Figure 6C) were all highly expressed in GBM. We further explored the associations among prognostic biomarkers. C5AR1 showed a strong correlation with IL10 (R = 0.75, *p* < 0.05; Figure 6D). There was a weak correlation between C5AR1 and PPP4C (R = −0.18, *p* = 0.025; Figure 6E). As shown in Figure 6F, there was no obvious relationship between IL10 and PPP4C (R = −0.15, *p* = 0.057; Figure 6F). These results indicate that the prognostic biomarkers might have a combined influence on the prognosis of GBM. Using the HPA database, we validated the translation expression level of the prognostic biomarkers with three GBM-HPA samples and three normal-HPA samples. We observed strong or moderate staining of the prognostic biomarkers: C5AR1, high staining (Figure 7B); IL10, high staining (Figure 7D); and PPP4C, medium staining (Figure 7F). These results imply that the translation levels of prognostic biomarkers were high in GBM samples. In the three normal samples, the prognostic biomarkers showed weak or no staining: C5AR1, no staining detected (Figure 7A); IL10, no staining detected (Figure 7C); and PPP4C, low staining (Figure 7E). We also found higher expression of C5AR1 (*p* < 0.05; Figure 8A), IL10 (*p* < 0.05; Figure 8B), and PPP4C (*p* < 0.05; Figure 8C) in GBM tissues than in normal peritumoral tissues by RT-qPCR.

### 3.6. Establishing an Immune-Related Prognostic Signature (IPS)

Following this, we built a risk-predicting system (immune-related prognostic signature) using all the prognostic biomarkers (C5AR1, IL10, and PPP4C) to characterize the risk of GBM patients. Risk scores for the GBM samples were calculated using the following formula: risk score = 0.194 * ExpC5AR1 + 0.046 * ExpIL10–0.665 * ExpPPP4C, which was determined by multivariate Cox regression analysis (Figure 5A). With this risk system, the data of 165 GBM patients from TCGA-GBM were divided into two groups (high-risk group, n = 82; low-risk group, n = 83) by setting the median risk score as the cut-off criterion. We found that GBM patients with lower risk scores had better OS than those with higher risk scores by survival analysis (*p* = 0.032; Figure 9A). Figure 9B shows the risk system’s ROC values (1-year AUC: 0.529; 3-year AUC: 0.685; 5-year AUC: 0.771). By examining the distribution between the two groups (Figure 9C), we discovered that patients in the high-risk group had a higher mortality rate than the low-risk group. We repeated the process with MMD2 to test the repeatability and accuracy of this signature. We still classified GBM into the high-risk (n = 55) and low-risk (n = 55) groups. GBM patients with higher risk scores had significantly worse OS (*p* = 0.0062; Figure 9D), in line with previous conclusions. The predicted values at 1, 3, and 5 years were accurately assessed by MMD2 as 0.639, 0.748, and 0.754, respectively (Figure 9E). The conclusions in Figure 9F are similar to those from the TCGA-GBM data.

### 3.7. Creation of Nomogram of Clinical Usefulness Depending on Immune-Related Prognostic Signature

To provide a visual prognosis prediction tool for clinicians, we attempted to construct a nomogram assessed by the IPS. We first obtained the risk score evaluated by the IPS and some essential clinical factors. Univariate Cox analysis indicated that risk score (*p* = 0.027) and additional pharmaceutical therapy (*p* = 0.014) were found to be strongly related to the OS of GBM patients (Figure 10A). Consequently, the risk score (*p* = 0.009) can then be used to reliably assess the prognosis of GBM patients, according to multivariate Cox analysis (Figure 10B). Then, based on the risk score and extra pharmaceutical therapy, we created a nomogram (Figure 10C). As concluded from the calibration curve, the nomogram could effectively predict the survival of GBM patients (Figure 10D–F), regardless of long-term mortality (5-year survival rate prediction; Figure 10F). DCA was also used to determine the nomogram’s net clinical benefit. The result suggests that when predicting the 1-year survival rate (0.35 < Pt < 0.50; Figure 10G), 3-year survival rate (0.65 < Pt < 0.80; Figure 10H), and 5-year survival rate (0.65 < Pt < 0.80; Figure 10I), the nomogram based on risk score was more efficient than one without risk score assessed by the IPS.

### 3.8. Identification of KEGG Signaling Pathways with Risk Signatures

In order to investigate the potential function of the IPS, we conducted GSEA. Based on the previously set standards, we concluded that the IPS plays a significant role in adhesion molecules (CAMS), chemokine signaling pathway, cytokine–cytokine receptor interaction, JAK STAT signaling pathway, natural killer cell-mediated cytotoxicity, T cell receptor signaling pathway, and Toll-like receptor signaling pathway. The detailed information of these pathways is shown in Figure 11A.

### 3.9. Correlation of IPS with Immune Infiltration Levels in GBM

We also wanted to see whether there was a relationship between our index and immune cells. As indicated in Figure 11B, this signature was significantly associated with B cells, (*p* < 0.05), macrophages (*p* < 0.01), myeloid dendritic cells (*p* < 0.01), neutrophils (*p* < 0.01), and CD4+ T cells (*p* < 0.01). We also explored the association between genes (used for IPS construction) and immune cell types. C5AR1 was positively associated with macrophages (*p* < 0.01), myeloid dendritic cells (*p* < 0.01), CD4+ T cells (*p* < 0.01), and neutrophils (*p* < 0.01). IL10 was positively correlated with B cells (*p* < 0.01), macrophages (*p* < 0.01), myeloid dendritic cells (*p* < 0.01), and neutrophils (*p* < 0.01). PPP4C was positively associated with macrophages (*p* < 0.01), myeloid dendritic cells (*p* < 0.01), CD4+ T cells (*p* < 0.01), and neutrophils (*p* < 0.01). We also validated the relationships between IPS and immune cell types by using CIBERSORT (Figure 11C), and the results were similar to those when using TIMER.

## 4. Discussion

Glioblastoma (GBM), with an annual incidence of 5.26/100,000 and 17,000 new diagnoses per year, is thought to be the most common primary malignant brain tumor. GBM often portends a poor quality of life and poor prognosis for patients [29]. The 5-year mortality rate exceeds 90%. Median survival of GBM is only 12.6 months [30]. The development of resection techniques, chemotherapy strategies, and radiation therapy for treating GBM has slowed. What is worse, the progress has not translated into significant improvements in patient survival [30]. Obviously, effective therapeutic targets and prognostic biomarkers are urgently needed in clinical practice. Thus, we attempted to identify novel prognostic biomarkers to provide new options for prognosis prediction and immunotherapy of GBM.

Immunotherapy is an important method to prevent and treat tumors, and it is a research hotspot [31]. Currently, more researchers are working on screening new prognostic biomarkers relevant to the immune microenvironment, but similar studies on GBM are still scarce. The purpose of this research is to find biomarkers related to the prognosis of GBM. For the first time, Hou et al. explored the correlation between GBM prognosis and immune-associated genes by using GSE4290, GSE50161, and GSE2223 [31]. They investigated 48 immune system genes that may influence GBM prognosis. However, there was a paucity of validation for those 48 genes. The study by Liang et al. confirmed the role of immune-related genes in predicting the GBM prognosis [32], but similarly, the problem with this study was the lack of validation of the prediction model. To learn the strengths and to avoid the weaknesses of these studies, we attempted to explore immune-related prognostic markers using several approaches based on multiple datasets and databases.

We performed not only DEG identification, as they did, but also WGCNA using IRGS collected from the ImmPort database. Then, we selected 29 overlapping genes between the results of the two analyses. Three IRGs (complement C5a receptor 1 (C5AR1), interleukin 10 (IL10), and protein phosphatase 4 catalytic subunit (PPP4C)) were further determined by conducting two kinds of survival analysis. In this way, we identified these three IRGs as potential biomarkers for predicting the prognosis of GBM and further validated them with different methods. TCGA-GBM data, GEPIA, and the HPA database showed that the three IRGs were highly expressed in GBM tissues compared to normal tissues. Previous studies concluded that C5AR1 was highly expressed in certain types of cancer. C5AR1 was also shown to induce breast cancer glycolysis by regulating m6A methylation [33]. Furthermore, C5AR1 was identified as a master regulator in colorectal tumorigenesis via immune modulation [34]. A study by Cheng et al. analyzed the immune-related gene set in glioblastoma and identified IL10 as one of the eight genes with the greatest prognostic value [35]. This result confirms the reliability of the results of the present study. It has been reported that the transcription level of PPP4C is higher in pancreatic cancer than in the normal pancreas and is associated with the pathogenesis and progression of pancreatic cancer [36]. Moreover, inhibiting the expression of PPP4C by siRNA can significantly inhibit the proliferation and migration of breast cancer cells [37]. PPP4C plays a prominent role in the progression of breast cancer and is used as a new biomarker to improve the accuracy of breast cancer diagnosis [37].

Altogether, this study verifies the notion that the three IRGs are closely related to tumor prognosis and might serve as new immune-related prognostic biomarkers in GBM. Furthermore, we found an IPS via these three prognostic biomarkers. It is worth mentioning that the IPS, which was relatively well-constructed (combining DEG and WGCNA), may be the first risk-prediction system based on multiple datasets with good clinical applicability. We created a nomogram using the prognostic signature and additional pharmaceutical therapy to turn the risk signature into a clinical reality. The IPS-based nomogram was instantly confirmed as a predictor of OS probability in GBM.

This study had certain limitations. As multi-dataset-based research, although this bioinformatics study was reasonably designed, it lacked external experimental validation. In future research, more experiments are needed to further clarify the molecular mechanism of prognostic biomarkers in GBM. In addition, we could collect clinical data and validate the prediction value of this IPS by using our own data in the near future.

## 5. Conclusions

In this study, three IRGs were discovered and confirmed to be novel immune-related prognostic biomarkers in GBM. The establishment of a prognostic signature to assess and predict GBM prognosis will facilitate the establishment of more effective immunotherapy strategies. At the same time, a nomogram was constructed based on the immune-related prognostic index, which was helpful in predicting the prognosis of GBM patients more intuitively.

## Figures and Tables

**Figure 1 cells-11-03000-f001:**
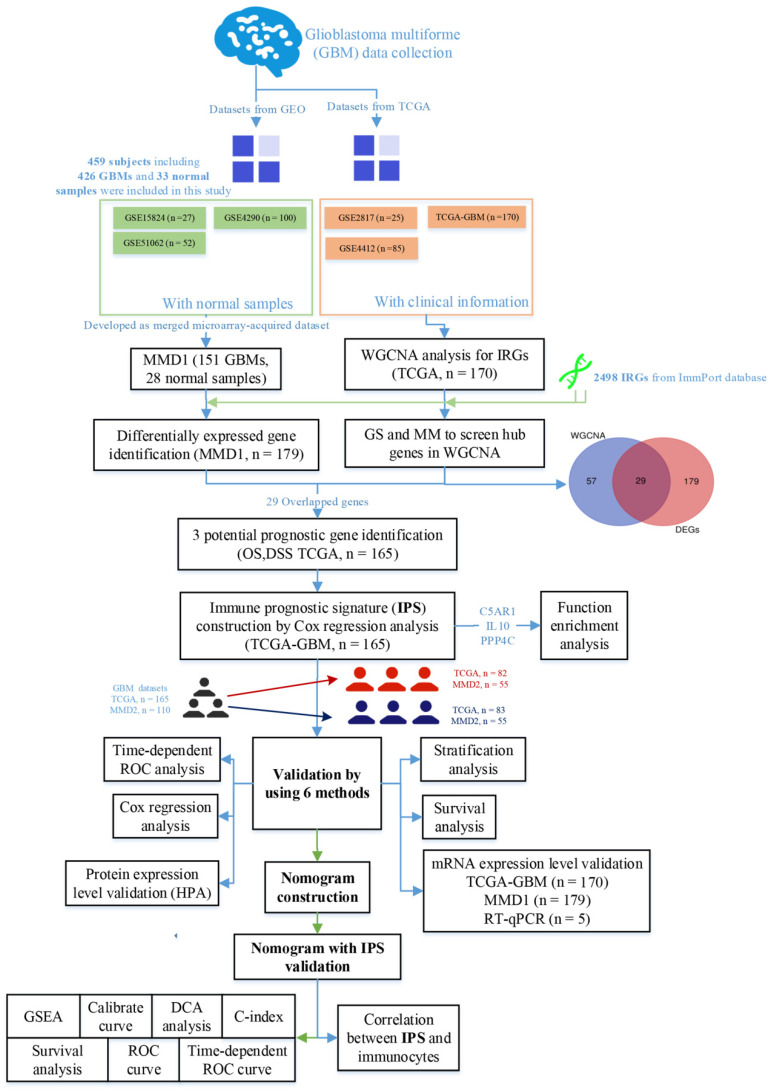
Flow diagram of this study demonstrating preparation, processing, and validation of data.

**Figure 2 cells-11-03000-f002:**
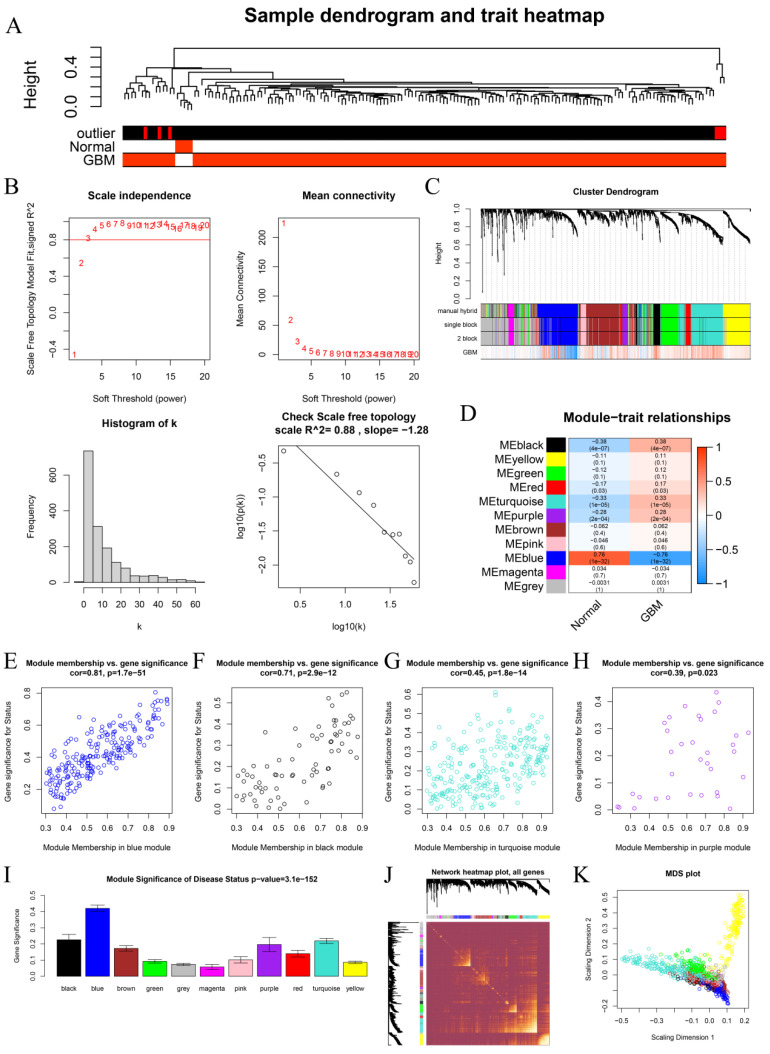
Associated module connected to clinical characteristic recognition. (**A**) Trait heatmap and sample dendrogram. Color intensity indicates patient condition. (**B**) Scale-free fit index analysis of soft-thresholding powers (β); mean connectivity index analysis of soft-thresholding powers; histogram of frequency distribution when β = 3; check scale-free topology when β = 3. (**C**) Dendrogram of clusters of differentially expressed genes based on dissimilarity measure (1-TOM). (**D**) Heatmap showing relationship between module eigengenes and ccRCC clinical data. (**E**–**H**) Scatter diagrams of gene module membership correlation in blue, black, turquoise, and purple modules with gene significance. (**I**) Average gene significance distribution in modules linked to GBM illness status. (**J**) Network heatmap plot for all genes in WGCNA. (**K**) Traditional MDS plot with TOM dissimilarity as its input. Module assigns color for each gene-designated dot.

**Figure 3 cells-11-03000-f003:**
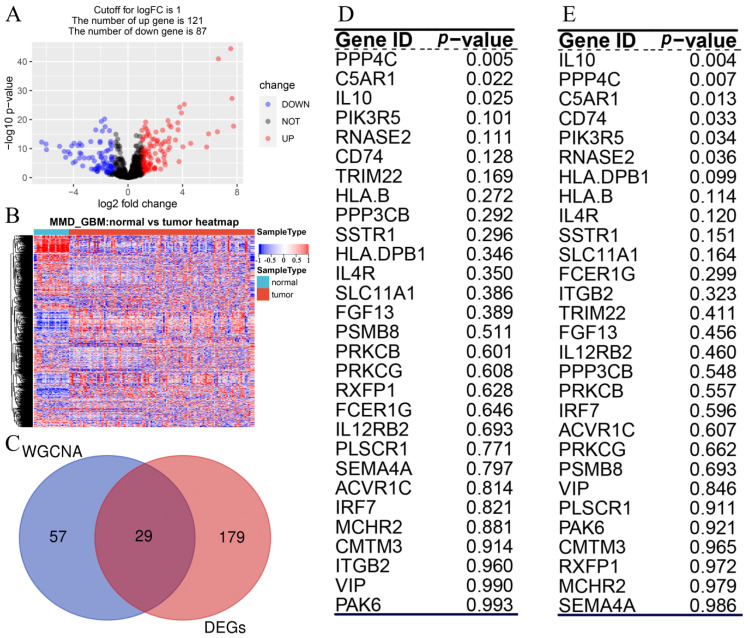
IRG identification. (**A**) Volcano plot displaying differentially expressed IRGs in MMD1-GBM data. (**B**) Heatmap of IRGs differently expressed in tumor samples compared to normal samples (fold change >1, *p* < 0.05, MMD1). (**C**) Venn diagrams of DEGs and IRGs identified in WGCNA. (**D**) Result of independent survival analysis for 29 IRGs in TCGA-GBM(OS). (**E**) Result of independent survival analysis for 29 IRGs in TCGA-GBM(DSS).

**Figure 4 cells-11-03000-f004:**
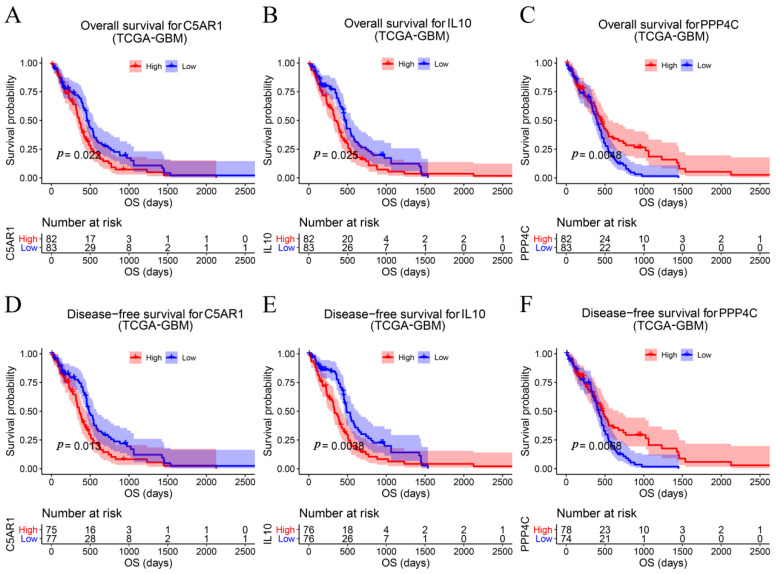
Verification of hub genes. Kaplan–Meier survival curve shows that GBM patients with higher hub gene expression had considerably lower overall survival—(**A**) C5AR1, (**B**) IL10, and (**C**) PPP4C—and disease-free survival—(**D**) C5AR1, (**E**) IL10, and (**F**) PPP4C.

**Figure 5 cells-11-03000-f005:**
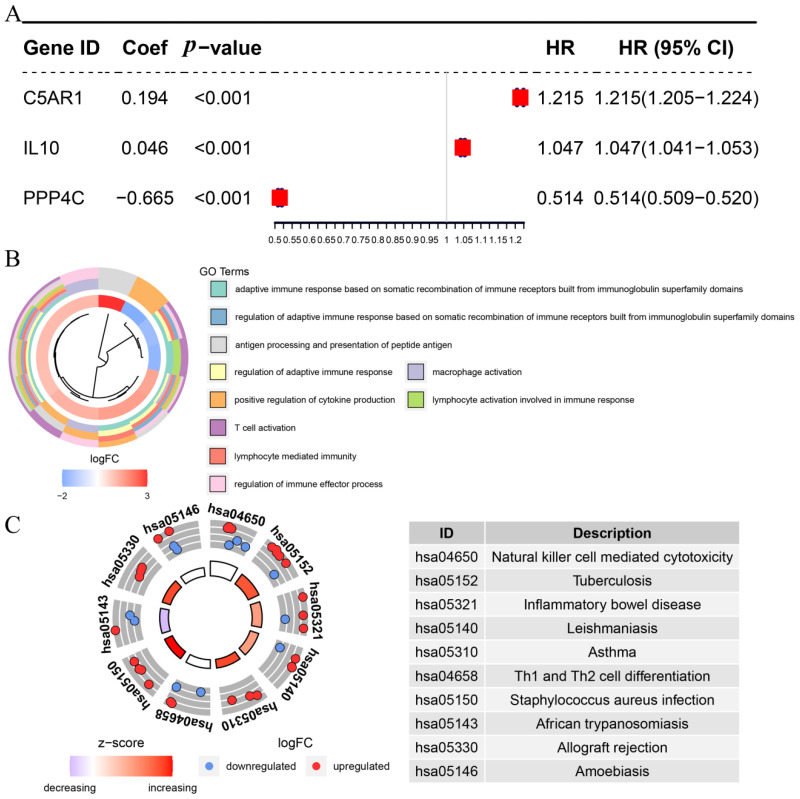
(**A**) Multivariate Cox analysis of three hub IRGs; (**B**) GO enrichment terms for hub IRGs. The inner ring is a bar plot where the color of the bar indicates the logFC of the term. The colors in the outer ring indicates the description of GO terms; (**C**) KEGG circle plot for IRGs. The inner ring is a bar plot where the height of the bar indicates the significance of the term. Red dots in the outer ring indicate upregulated genes and blue dots indicate downregulated genes.

**Figure 6 cells-11-03000-f006:**
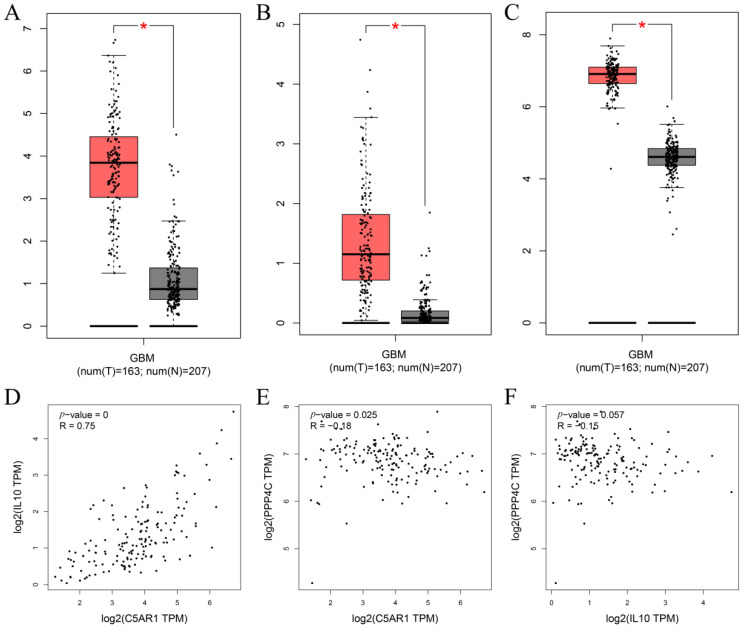
Expression of (**A**) C5AR1, (**B**) IL10, and (**C**) PPP4C in GBM was markedly higher than in normal tissues from TCGA-GBM database (* *p* < 0.05). Relationship analysis of three key genes: (**D**) C5AR1 and IL-10, (**E**) C5AR1 and PPP4C, and (**F**) PPP4C and IL-10.

**Figure 7 cells-11-03000-f007:**
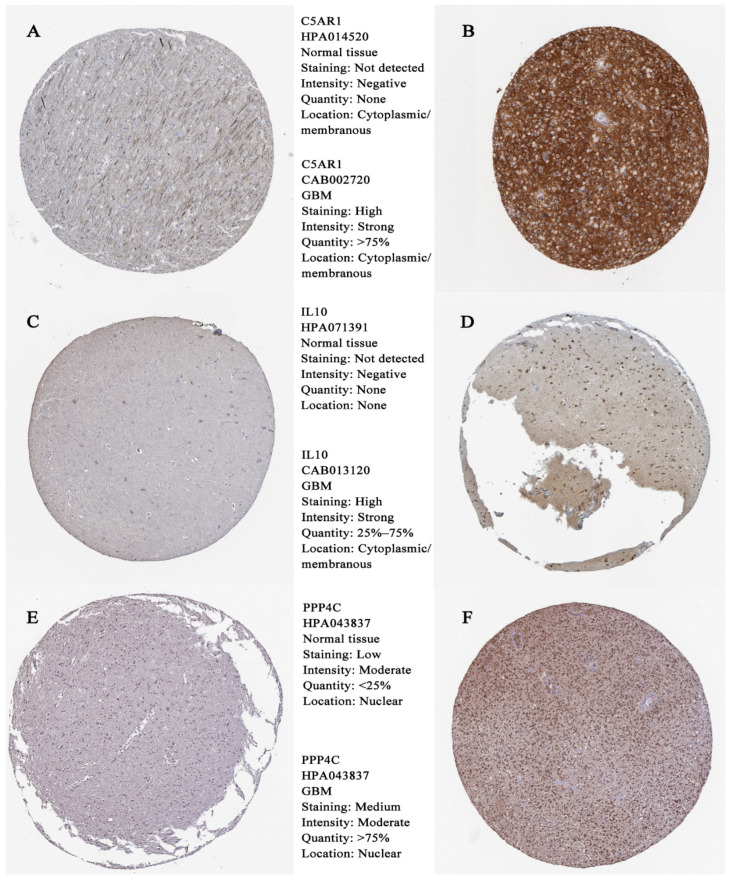
Hub gene expression in normal human and GBM tissue detected by immunochemistry. (**A**) Representative IHC staining of C5AR1 in normal human tissue; (**B**) IHC staining typical for C5AR1 in human GBM tissue; (**C**) IHC staining typical for IL10 in normal human tissue; (**D**) IHC staining typical for IL10 in human GBM tissue; (**E**) IHC staining typical for PPP4C in normal human tissue; (**F**) IHC staining typical for PPP4C in human GBM tissue.

**Figure 8 cells-11-03000-f008:**
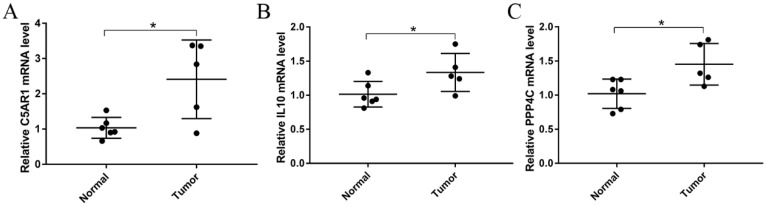
Relative mRNA levels of (**A**) C5AR1, (**B**) IL10, and (**C**) PPP4C in GBM (n = 5) were significantly higher than in normal peritumoral tissues (n = 6) (Student’s *t*-test, * *p* < 0.05).

**Figure 9 cells-11-03000-f009:**
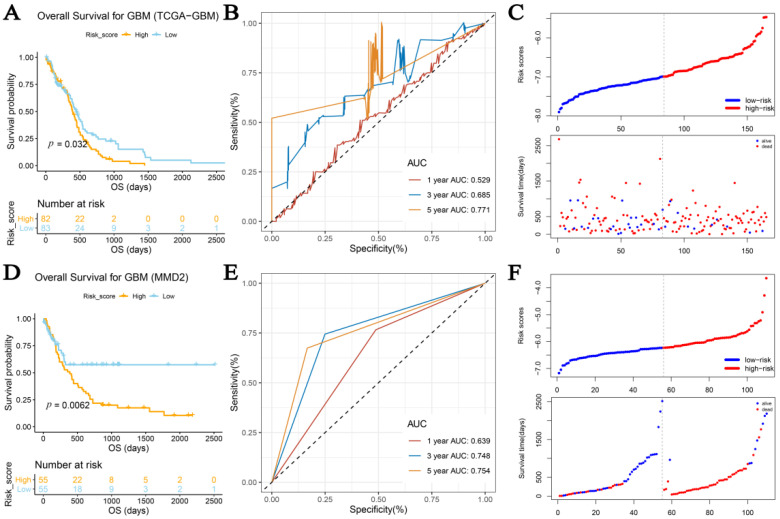
Cox regression analysis and construction of new immune-related prognostic signature. (**A**) Based on TCGA-GBM data, graphs show Kaplan–Meier OS curves for high- and low-risk groups. (**B**) ROC curve shows accuracy of immune-related prognostic signature for OS. (**C**) Distribution of risk scores among GBM patients according to TCGA-GBM data; number of survivors (red) and non-survivors (blue) with various risk scores according to MMSD. (**D**) Kaplan–Meier OS curves for high- and low-risk groups using MMD2 data. (**E**) ROC curves showing precision of immune-related prognostic signature for OS. (**F**) Distribution of GBM patient risk ratings; proportion of survivors and non-survivors with various risk ratings depending on MMSD.

**Figure 10 cells-11-03000-f010:**
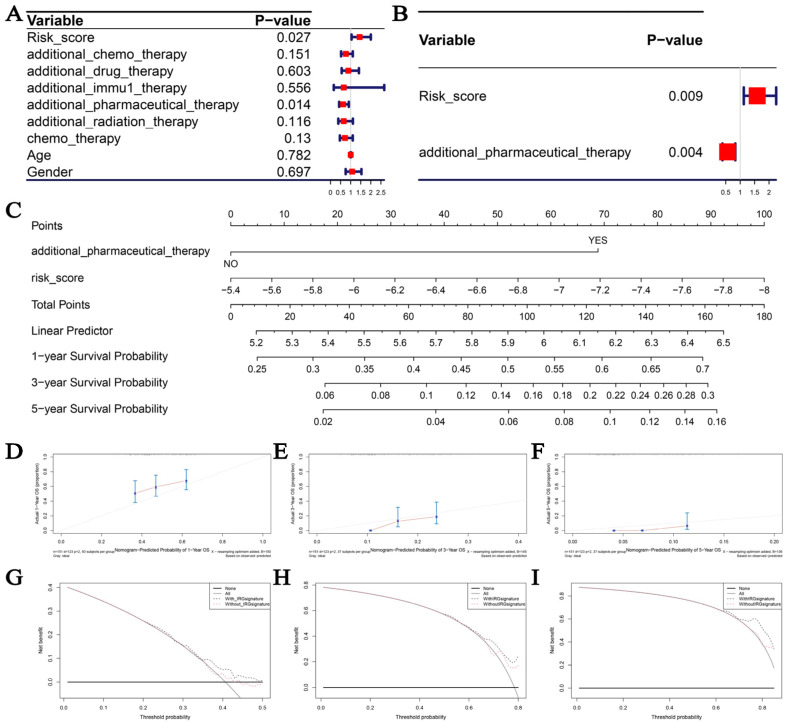
(**A**) Forest plot summary of OS univariate analysis of risk score, chemotherapy, drug therapy, immunotherapy, pharmaceutical therapy, radiation therapy, age, and gender based on TCGA-GBM data. (**B**) OS univariate analysis of risk score via forest plot summary, risk score, and pharmaceutical therapy based on TCGA-GBM data. (**C**) Immune-related prognostic signature-based nomogram for estimating percentage of patients with 1-, 3-, or 5-year OS. Scales are used to indicate the range of values for each variable on the corresponding line segment, and the length of the line segment reflects the contribution of this factor to the outcome event. (**D**–**F**) Calibration charts for predicting 1-, 3-, and 5-year OS, respectively. (**G**–**I**) DCA for assessment of clinical utility of immune-related prognostic signature for 1-, 3-, and 5-year OS; percent of threshold probability is shown on x-axis, net benefit is shown on y-axis.

**Figure 11 cells-11-03000-f011:**
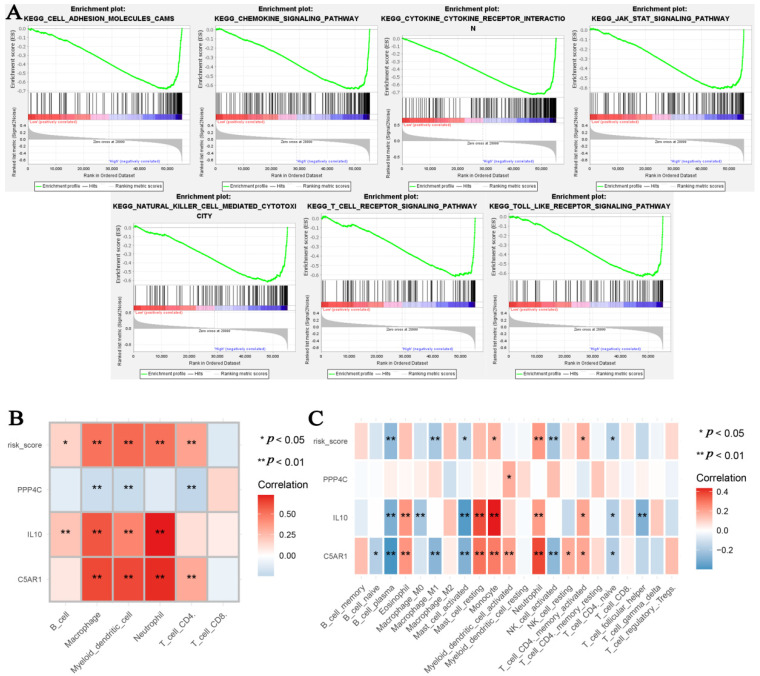
(**A**) Genome enrichment analysis (GSEA) of immune-related prognostic signature. (**B**) Associations between risk score and six types of immune cells found in TIMER. (**C**) Associations between risk score and 22 immune cell behaviors found in CIBERSORT. * *p* < 0.05, ** *p* < 0.01.

## Data Availability

The accession numbers of all the databases analyzed for this study are included in the article.

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
