# Peer review of "Identification of an Immune-Related Prognostic Signature for Glioblastoma by Comprehensive Bioinformatics and Experimental Analyses"

_cells, 2022, doi:10.3390/cells11193000_

Round 1

Reviewer 1 Report

1.   Such a data mining study on TCGA for GBM prognostic markers is not novel (Di Jia, 2018), and narrowing down the gene pool from all the genes to IRGs does not provide more values in downstream analyses. Instead, they might decrease the predictive values. The authors should comment on this or provide evidence that starting by using all genes won’t offer more prognostic values, e.g., higher ROCs.

2.     It is unclear why the authors didn’t perform DEG analyses on the TCGA data screening for IPS. Understandably, WGCNA was performed only using TCGA data as it was followed by studies using clinical features. Still, I think the authors could have completed the same DEG analyses on TCGA as on MMD1 data. Would it change the final set of IPS genes if that was done?

3.   Line 161 and 328, it is unclear how the weights (coefficients) of the Risk score equation were determined. If the authors manually set the coefficients, please justify.

4.   The paper itself is tough to understand or follow. It has countless grammar mistakes and typos. Below are some of the errors (not all) that could be easily identified from the manuscript:

a.     Line 81, “data normalization (normalization and log2 transformation)”. I suppose here you mean “library-size normalization and log-transformation.”

b.     Line 108, what does this mean:  “we obtained 170 IRGs” after declaring 1617 IRGs in line 104?

c.     Line 126, what is the threshold for fold change?  “Log2fc (124) >= 1” must be a typo. Since this is an important parameter, it is critical and could directly change your results.

d.     Line 244, “88 genes with were identified”.

e.     Line 257, “positive P values.”

f.      Line 331, “patients who risk scores.”

g.     Word “contained” was misused in many places in the manuscript. Please change it.

5.     Captions for Figure 3, panel C-E are wrong or misleading.

6.     Figure 7 is tough to understand. The caption is misleading, and its order is painful to read.

Author Response

Dear Reviewer,

Thanks very much for taking your time to review this manuscript. I really appreciate all your comments and suggestions! Please find my itemized responses in below and my revisions/corrections in the re-submitted files.

Thanks again!

Comment1:  Such a data mining study on TCGA for GBM prognostic markers is not novel (Di Jia, 2018), and narrowing down the gene pool from all the genes to IRGs does not provide more values in downstream analyses. Instead, they might decrease the predictive values. The authors should comment on this or provide evidence that starting by using all genes won’t offer more prognostic values, e.g., higher ROCs.

response: Immunotherapy has shown strong antitumor activity in the treatment of a variety of tumors, such as melanoma, non-small cell lung cancer, kidney cancer, prostate cancer and other solid tumors. Many tumor immunotherapy drugs have been approved for clinical application by the food and Drug Administration (FDA) of the United States. However, immunotherapy has made little progress in glioblastoma over the past 10 years because of the intracranial location and heterogeneity of tumors, and the distinct immunosuppressive tumor microenvironment. New prognostic biomarkers should be developed. We use IRGs as prognostic markers rather than all genes because we want to explore immune-related prognostic markers of GBM and their possible mechanisms, providing a basis for individualized diagnosis and treatment. At the same time, our signature has the same reliable ROC as other signatures.

Comment2:  It is unclear why the authors didn’t perform DEG analyses on the TCGA data screening for IPS. Understandably, WGCNA was performed only using TCGA data as it was followed by studies using clinical features. Still, I think the authors could have completed the same DEG analyses on TCGA as on MMD1 data. Would it change the final set of IPS genes if that was done?

response: DEG analyses and WGCNA both screen genes based on the expression difference between normal tissues and tumor tissues. We completed the DEG analyses on MMD1 data because we want to screen some genes which provide values in both TCGA and MMD1. 

Comment3: Line 161 and 328, it is unclear how the weights (coefficients) of the Risk score equation were determined. If the authors manually set the coefficients, please justify.

response: We obtained the weights(coefficients) by multivariate cox regression analysis, and we revised it in the new manuscript.

Comment4: The paper itself is tough to understand or follow. It has countless grammar mistakes and typos. Below are some of the errors (not all) that could be easily identified from the manuscript:

  1. Line 81, “data normalization (normalization and log2 transformation)”. I suppose here you mean “library-size normalization and log-transformation.”
  2. Line 108, what does this mean:  “we obtained 170 IRGs” after declaring 1617 IRGs in line 104?
  3. Line 126, what is the threshold for fold change?  “Log2fc (124) >= 1” must be a typo. Since this is an important parameter, it is critical and could directly change your results.
  4. Line 244, “88 genes with were identified”.
  5. Line 257, “positive P values.”
  6. Line 331, “patients who risk scores.”
  7. Word “contained” was misused in many places in the manuscript. Please change it.

response: Sorry for the above grammar mistakes and typos, we had revised these errors in our new manuscript. And we submitted our manuscript to MDPI for English editing.

Comment5: Captions for Figure 3, panel C-E are wrong or misleading.

response:We have revised the captions for Figure 3, panel C-E as follows:

(C) Venn diagrams of DEGs and IRGs identified in WGCNA.

(D) Result of independent survival analysis for 29 IRGs in TCGA-GBM(OS).

(E) Result of independent survival analysis for 29 IRGs in TCGA-GBM(DSS).

Comment6: Figure 7 is tough to understand. The caption is misleading, and its order is painful to read

response: :We have revised the caption for Figure 7 as follows:

Hub gene expression in normal human and GBM tissue detected by immunochemistry. (A) Representative IHC staining of C5AR1 in normal human tissue; (B) IHC staining typical for C5AR1 in human GBM tissue; (C) IHC staining typical for IL10 in normal human tissue; (D) IHC staining typical for IL10 in human GBM tissue; (E) IHC staining typical for PPP4C in normal human tissue; (F) IHC staining typical for PPP4C in human GBM tissue

Reviewer 2 Report

Glioblastoma (GBM) is a common type of brain tumor with poor prognosis. The prognostic indicators for glioblastoma remain largely unknown. In this manuscript, Ye et al., reported bioinformatic analyses and experimental validation of immune-related prognostic signature (IPS) genes in GBM. They highlighted three immune-related genes PPP4C, C5AR1 and IL-10, based on which they constructed a nomogram and proposed that it had better accuracy than the one without the IPS genes. In general, the study has clinical significance and was reasonably planned. The authors applied a comprehensive set of bioinformatic tools, and the analyses were done in a scientific fashion. The authors are required to put more effort in refining their data presentation and written English. The specific comments towards the current version of the manuscript are listed below.

11)    Overall, there are quite a few grammatical and vocabulary errors throughout the manuscript. I suggest that the manuscript is further edited by a native speaker.

22)    In the introduction part, the authors need to further elaborate on the current advances in the immunotherapy of glioblastomas. This is important for the readers to understand why the authors focus on immune-related gene signatures.

33)    Figure 3B, the color of the heatmap looks weird (there’s no blue color in it). I suggest the authors use blue scale for all negative values.

44)    Figure 4, I suggest the authors add gene name to the title of each plot, rather than putting them in the legend.

55)    Figure 5B and C, these circos plots are somewhat confusing and hard to follow. The figure legends for these figures are too brief and not informative. For example, what does each dot represent in figure 5c? I suggest that the authors use more reader-friendly ways to present their data.

66)    The authors need to further elaborate on the results of figure 10. This figure is hard to follow and understand. The description and legend are too brief.

77)    Please make sure the qPCR primer and IHC antibody information is available.

Author Response

Dear Reviewer,

Thanks very much for taking your time to review this manuscript. I really appreciate all your comments and suggestions! Please find my itemized responses in below and my revisions/corrections in the re-submitted files.

Thanks again!

Comment1: Overall, there are quite a few grammatical and vocabulary errors throughout the manuscript. I suggest that the manuscript is further edited by a native speaker.

Response:Thanks for the advice,we had submitted the manuscript to MDPI for English editing and corrected these errors.

Comment2: In the introduction part, the authors need to further elaborate on the current advances in the immunotherapy of glioblastomas. This is important for the readers to understand why the authors focus on immune-related gene signatures.

Response:  We have improved the language quality of the introduction. and added elaboration on the current advances in glioblastoma immunotherapy as follows.

However, immunotherapy has made little progress in glioblastoma over the past 10 years because of the intracranial location and heterogeneity of tumors, and the distinct immunosuppressive tumor microenvironment. New prognostic biomarkers should be developed .

Comment3: Figure 3B, the color of the heatmap looks weird (there’s no blue color in it). I suggest the authors use blue scale for all negative values

Response: We have change the color of the heatmap and use blue scale for all negative values.

Comment4:  Figure 4, I suggest the authors add gene name to the title of each plot, rather than putting them in the legend.

 Response: We have added gene name to the title of each plot.

Comment5:  Figure 5B and C, these circos plots are somewhat confusing and hard to follow. The figure legends for these figures are too brief and not informative. For example, what does each dot represent in figure 5c? I suggest that the authors use more reader-friendly ways to present their data.

 Response: We have revised figure legends as follows:

Figure 5. (A) Multivariate Cox analysis of three hub IRGs; (B) GO enrichment terms for hub IRGs. The inner ring is a bar plot where the color of the bar indicates the logFC of the term. The colors in the outer ring indicates the description of GO terms; (C) KEGG circle plot for IRGs. The inner ring is a bar plot where the height of the bar indicates the significance of the term. Red dots in the outer ring indicate upregulated genes and blue dots indicate downregulated genes.

Comment6:  The authors need to further elaborate on the results of figure 10. This figure is hard to follow and understand. The description and legend are too brief

Response: We have revised figure legends as follows:

Figure 10. (A) Forest plot summary of OS univariate analysis of risk score, chemotherapy, drug therapy, immunotherapy, pharmaceutical therapy, radiation therapy, age, and gender based on TCGA-GBM data. (B) OS munivariate analysis of risk score via forest plot summary, risk score, and pharmaceutical therapy based on TCGA-GBM data. (C) Immune-related prognostic signature-based nomogram for estimating percentage of patients with 1-, 3-, or 5-year OS. Scales are used to indicate the range of values for each variable on the corresponding line segment, and the length of the line segment reflects the contribution of this factor to the outcome event. (D–F) Calibration charts for predicting 1-, 3-, and 5-year OS, respectively. (G–I) DCA for assessment of clinical utility of immune-related prognostic signature for 1-, 3-, and 5-year OS; percent of threshold probability is shown on x-axis, net benefit is shown on y-axis.

Comment7:  Please make sure the qPCR primer and IHC antibody information is available.

Response: Our qPCR primer is available in supplementary materials, and  our IHC staining acquired from the Human Protein Atlas (HPA) database.

Reviewer 3 Report

Dear authors,

in the MS you apply bioinformatics methods (WGCNA and DEG) to Glioblastoma datasets to identify immune related gene signatures and validate the ientified hub gene in tissue specimen.

Overall the design and execution of the study is very well done, with a few minor things:

For example, Line 77: Microarrays have intensity values and not counts. That is for RNASeq data.

You should screen the MS for milar little glitches.

The introduction, however is a bit short and should be improved

Author Response

Dear  Reviewer,

Thanks very much for taking your time to review this manuscript. I really appreciate all your comments and suggestions! Please find my itemized responses in below and my revisions/corrections in the re-submitted files.

Thanks again!

1.For example, Line 77: Microarrays have intensity values and not counts. That is for RNASeq data.

I have corrected milar glitches and the manuscript had been submitted to MDPI for English editing.

2.The introduction, however is a bit short and should be improved

We have improved the language quality of the introduction. and added elaboration on recent advances in glioblastoma immunotherapy asfollows.

However, immunotherapy has made little progress in glioblastoma over the past 10 years because of the intracranial location and heterogeneity of tumors, and the distinct immunosuppressive tumor microenvironment. New prognostic biomarkers should be developed [6]. 

Round 2

Reviewer 1 Report

The authors have addressed all my comments. The writing also improved a lot. However, some typos and formatting issues still needed to be further checked. Below is a few of them:

1. Line 24: 'T Three immune.'

2. Line 53: 'urgent required' should be changed to 'urgently required.'

3. Line 236: 'were classed' should be changed to 'were classified.'

4. Some of the revised sentences were written in Arial format, while the rest of the manuscript was in Times. Please make it concordant.

Reviewer 2 Report

The manuscript has been improved and I don’t have further comments.